# Micropropagation of Rare *Scutellaria havanensis* Jacq. and Preliminary Studies on Antioxidant Capacity and Anti-Cancer Potential

**DOI:** 10.3390/molecules26195813

**Published:** 2021-09-25

**Authors:** Lani Irvin, Yarelia Zavala Ortiz, Kamila Rivera Rivera, Brajesh Nanda Vaidya, Samantha H Sherman, Rosalinda Aybar Batista, Juan A. Negrón Berríos, Nirmal Joshee, Alok Arun

**Affiliations:** 1Agricultural Research Station, Fort Valley State University, Fort Valley, GA 31030, USA; lani.irvin@unco.edu (L.I.); braz_v@hotmail.com (B.N.V.); ssherma3@wildcat.fvsu.edu (S.H.S.); Josheen@fvsu.edu (N.J.); 2Institute of Sustainable Biotechnology, Inter American University of Puerto Rico, Barranquitas, PR 00794, USA; yarzav6165@br.uipr.edu (Y.Z.O.); karivera1343@gmail.com (K.R.R.); raybar@br.inter.edu (R.A.B.); janegron@br.inter.edu (J.A.N.B.)

**Keywords:** conservation, HCT 116 cells, polyphenol, trichomes

## Abstract

We report the development of in vitro propagation protocols through an adventitious shoot induction pathway for a rare and medicinal *Scutellaria havanensis.* In vitro propagation studies using nodal explants showed MS medium supplemented with 10 µM 6-Benzylaminopurine induced the highest number of adventitious shoots in a time-dependent manner. A ten-day incubation was optimum for shoot bud induction as longer exposures resulted in hyperhydricity of the explants and shoots induced. We also report preliminary evidence of *Agrobacterium tumefaciens* EHA105-mediated gene transfer transiently expressing the green fluorescent protein in this species. Transformation studies exhibited amenability of various explant tissues, internode being the most receptive. As the plant has medicinal value, research was carried out to evaluate its potential antioxidant capacity and the efficacy of methanolic leaf extracts in curbing the viability of human colorectal cancer cell line HCT116. Comparative total polyphenol and flavonoid content measurement of fresh and air-dried leaf extract revealed that the fresh leaf extracts contain higher total polyphenol and flavonoid content. The HCT 116 cell viability was assessed by colorimetric assay using a 3-(4,5-dimethyl-thiazol-2-yl)-2,5-diphenyltetrazolium bromide, showed a steady growth inhibition after 24 h of incubation. Scanning electron microscopy of leaf surface revealed a high density of glandular and non-glandular trichomes. This research provides a basis for the conservation of this rare plant and future phytochemical screening and clinical research.

## 1. Introduction

The genus *Scutellaria* (commonly skullcap or scullcap) belongs to the plant family Lamiaceae (mint family). The genus is widely distributed and represented by about 400 species globally [1]. More than 295 bioactive compounds have been isolated and identified in 35 species of *Scutellaria* [2]. Pharmacological studies have confirmed that total extracts or isolated flavonoids from *Scutellaria* species possess anti-lipoperoxidation, anti-platelet, anti-inflammatory [3], antitumor [4,5,6], anticancer [7], neuroprotective [8] antithrombotic, antioxidant [9,10], hepatoprotective, antibacterial [11], and antiviral [2] activities. A recent review details ethnobotanical knowledge surrounding various Scutellaria species from different parts of the world and current ongoing biomedical studies [12]. Extracts from various parts of *Scutellaria* species possess antitumor, hepatoprotective, antioxidant, antibacterial, and antiviral effects [13]. Baicalein, a phenolic flavonoid compound derived mainly from the root of *S. baicalensis*, resulted in a concentration-dependent inhibition of cell growth and induction of apoptotic cell death in human colon cancer cells [14,15]. Similarly, extracts of *S. barbata* have shown in vivo growth inhibitory effects in a number of cancers [16]. Our recent studies indicate that methanolic leaf extracts of *Scutellaria ocmulgee* (SocL) have a potential for developing neoadjuvant therapy for metastatic NSCLC (non-small cell lung cancer) cells employing a zebrafish model [17].

*Scutellaria havanensis* Jacq. (Havana skullcap) is a rare species based on its restricted distribution. *S. havanensis* grows as a groundcover herb on limestone rocks and cliffs, at lower to middle elevations in Puerto Rico, Mona Island, and the Bahamas whereas in Florida, Cuba, and Hispaniola it grows in rocky pinelands [18]. It has been used to treat several disorders, including those related to the central nervous system. The plant has been used for the treatment of scabies and as a diaphoretic and a febrifuge as an ethnomedicine [19]. Seeds are used against psoriasis, and sarcoptic mange [20]. Phytochemical screening of extracts from aerial parts of *S. havanensis* reported the presence of flavonoids, coumarins, triterpenoids, steroids, alkaloids, free amine groups, sugars, quinones, resins, and saponins [21]. In addition, twenty-five volatile compounds from the extract of aerial parts of *S. havanensis* were identified, β-caryophyllene (75.6%) being the most prominent compound [21]. Studies on *Scutellaria* extracts have shown its role in oxidative stress associated with several human pathogeneses [22,23,24]. The methanol and chloroform extracts of *S. havanensis* leaves and stems contain high concentrations of flavonoids including bioactive wogonin. Wogonin in the chloroform extract indicated its specific anti-plasmodial effect with respect to mammalian cells and thus could be useful for the development of antimalarial alternatives [25]. These compounds are synthesized in glandular trichomes that serve the function of defense and source of valuable chemicals and are also used as taxonomic tools. Type of trichomes and their location and frequency on the various parts of a plant can be used to check the secondary metabolite potential of a species/variety and can also assist in identifying adulterants in herbal biomass [26].

As this is a rare plant, an efficient micropropagation protocol will assist conservation and further research on biotransformation for bioactive molecules. There are successful reports on the rapid multiplication and genetic transformation of many Scutellaria species [27,28,29,30,31]. In a recent review article, the role and importance of transgenic studies have been emphasized in various medicinal plants [32]. It is evident from literature that the success of micropropagation is primarily dependent on the selection of explant and plant growth regulators used in the culture medium. As the secondary metabolite synthesis in medicinal plants is carried out in specialized structures called trichomes, fluorescent staining of secretory cells and structural details using a scanning electron microscope (SEM) are also presented. We also report evaluation of antioxidant capacity in fresh and air-dried leaf extracts which could be an important factor contributing to medicinal activity. We report preliminary results on the efficacy of methanolic extracts of *S. havanensis* leaves on steady growth inhibition of HCT 116 human colon cancer cells, as measured by MTT colorimetric assay.

## 2. Materials and Methods

### 2.1. Micropropagation

**Plant material, explant, and media preparation:***S. havanensis* plants were raised from seeds (Plant World Seeds, Newton Abbot, UK) in the greenhouse. Vigorously growing shoots were harvested and sterilized as reported earlier for *Scutellaria ocmulgee* [33] to initiate mother stock cultures to supply clean explants for experiments. Using single and two-node sterilized explants, mother stock plants were raised in test tubes on MS [34] basal medium (PhytoTechnology, Lenexa, KS, USA). Stock cultures were incubated in the culture room maintained at 25 ± 2 °C and 60% relative humidity with 16 h photoperiod provided by cool white fluorescent lamps. To initiate shoot induction experiments, one node explants were obtained from in vitro grown stock plants. Shoot induction experiment comprised of a control and four (0.05, 0.5, 5.0, and 10 µM) cytokinin 6-benzylaminopurine (BAP) (PhytoTechnology, Lenexa, KS, USA) treatments. Prior to autoclaving the shoot induction medium, 30 g L^−1^ sucrose and 4 g L^−1^ gelzan (PhytoTechnology, Lenexa, KS, USA) was added, and pH was adjusted to 5.8–5.9. Autoclaving was carried out for 20 min at 121 °C and 15 psi. After sterilization, 12 mL of shoot induction medium was dispensed into Petri dishes (60 mm × 15 mm; Thermo Fisher Scientific, Waltham, MA, USA). Four nodal explants were placed in a Petri dish for each BAP treatment and a total of five Petri dishes were inoculated per treatment. Cultures were incubated at 25 ± 2 °C with a 16 h photoperiod. To understand optimum exposure of BAP required for shoot induction, explants were transferred to MS basal elongation medium from various treatments after 0, 3, 5, 7, 10, and 14 days. Four explants from each BAP treatment were transferred and induced buds were allowed to elongate on MS basal medium for 21 days. The number of elongated adventitious shoots were counted after 21 days and statistical analysis was carried out. Culture responses were observed using a microscope (Leica EZ4W, Lenexa, KS, USA) and pictures were taken. Elongated shoots ranging 3–4 cm in length with 3–4 nodes were cut and transferred to MS medium supplemented with 5 µM IBA (índole butyric acid) for rooting. Rooted plantlets were acclimatized as per our lab protocol standardized for Scutellaria species [30,33] and then transferred to the greenhouse in pots for further growth, flowering, and seed set.

***Agrobacterium tumefaciens* mediated genetic transformation:** Genetic transformation studies on *S. havanensis* were carried out [33] with minor modifications. *Agrobacterium tumefaciens* EHA105 with binary plasmid pq35SGR [35] was tested for agroinfection and transformation efficiency. This study was carried out to test agroinfection and regeneration potential of various explants (node, internode, and petiole) on the basis of green fluorescent protein expression. Agroinfection was carried out by submerging explants in the bacterial suspension (OD_600_ of 0.2, 0.3, 0.4, 0.6, and 1.0) for 30 min with an occasional swirl. Fifteen replicates for each type of explant were used at each bacterial density point. Explants were wounded by scratching on the surface to facilitate agroinfection. After agroinfection, explants were dabbed on sterile filter paper to remove excess bacteria and incubated for co-cultivation in the dark for 72 h at 26–28 °C on moist sterile filter papers. Explants were checked for transient GFP expression using an Olympus microscope (BX 43, Olympus, Center Valley, PA, USA) equipped with epifluorescence illumination and a camera (DP 72, Olympus, USA). After completion of co-cultivation period, explants with positive GFP expression were rinsed with two changes of washing solution (liquid MS containing 250 mg L^−1^ of antibiotics carbenicillin and cefotaxime) (Phytotechnology Lab, Livonia, MI, USA) for 30 min each time. After washing, explants were gently dried on sterile filter paper by wicking away excess fluid and transferred to a selection plate that contained shoot induction medium (MS + 5 µM BAP) supplemented with 250 mg L^−1^ carbenicillin and cefotaxime both. After few changes of selection plates, explants with positive GFP expression were transferred to MS + 5 µM BAP + 100 mg L^−1^ carbenicillin + 100 mg L^−1^ cefotaxime for regeneration. Explants with positive GFP expression were photographed at this point and plant regeneration work is in progress.

**Fluorescent microscopy for secondary metabolite in trichomes:** For the detection of the secretory nature of trichomes, a natural product reagent was prepared [36] with a 5% aqueous solution of aluminum chloride (Fisher Scientific, Waltham, MA, USA) and 0.05% diphenyl boric acid-β-ethylaminoester (DPBA) (Sigma Aldrich, St. Louis, MO, USA) in 10% methanol (Burdick and Jackson, Muskegon, MI, USA). Leaves from greenhouse-grown and tissue cultured plants were soaked in the stain for 5–10 min in dark and observed under a fluorescent microscope (Olympus BX 43, Olympus, USA) equipped with a UV light source (X-Cite series 120 Q; Lumen Dynamics, Waltham, MA, USA) and DP 72 camera. Secondary metabolites in the trichomes on the abaxial and adaxial leaf surfaces were observed using a DAPI filter (with excitation light at λ 386 nm; emission λ 490 nm) under a fluorescent microscope. Trichomes fluoresce with different shades of yellow or light green depending on the type of compound present. A yellow–orange fluorescence serves as the identification of phenols as flavone compounds [37]. Auto-fluorescence is diagnostic of flavonoids and depending on the structure, will fluoresce dark yellow, green, or blue under UV-365 nm light [38].

**Scanning electron microscopy (SEM):** The leaf samples were prepared for SEM as optimized in our lab [33]. The images were captured using variable pressure-scanning electron microscope Hitachi 3400 NII (Hitachi High Technologies America, Inc., Chatsworth, CA, USA) at accelerating voltage of 10 kV and a working distance of 10 mm at Center for Ultrastructural Research (CURE), Agricultural Research Station, Fort Valley State University, GA.

### 2.2. Preparation of Leaf Extracts

Two samples of leaf tissues were randomly collected from mother plants, and cleaned, and weighed (2 g each). One sample was used for fresh extraction while the second sample was dried at room temperature (35 °C) in the dark for 7 days, weighed, and then used for extraction. For extraction, the leaves were homogenized with liquid nitrogen in a chilled (−20 °C) mortar and pestle. The homogenized leaf powder was transferred to a 125 mL Erlenmeyer flask containing 50 mL HPLC grade 100% methanol (Sigma-Aldrich, St. Louis, MO, USA). The flasks were left overnight (18 h at 28.5 °C) in the dark on an orbital shaker at 200 RPM (Labline 3508 Dual Action Shaker; Marshall Scientific, Hampton, NH, USA). The suspension was then transferred to 50 mL Falcon tubes (BD, Swedesboro, NJ, USA) to be centrifuged at 4000 RPM (5810 R; Eppendorf, Enfield, CT, USA) at 25 °C for 40 min. The supernatant was collected and the remaining pellet was extracted again for 1 h, with 25 mL of methanol. After the second extraction, the two extracts were combined and the pellets were discarded. The combined extract was filtered through a double layer of Whatman™ filter paper No. 41 (GE Healthcare Life Sciences, Stevenage, UK) and stored in air tight 50 mL Falcon tubes at 4 °C in the dark, until further analysis.

### 2.3. Biochemical Assays

Total polyphenol (TPP) content was determined by the Folin-Ciocalteu reagent (Sigma-Aldrich, MO, USA) method [39] and as modified [40,41] for the herbs in Lamiaceae family. Gallic acid (3,4,5-trihydroxybenzoic acid) was used to develop a standard curve [42]. A Multiskan GO Microplate Spectrophotometer (Thermo Fisher Scientific, USA) was used to read the extract solution absorbance at 765 nm. For each sample, five replicates were measured at 20 s intervals in three different runs. Total phenolic content was expressed as mg gallic acid equivalent/g dry or fresh extract (GAE mg/g).

Total flavonoid content was evaluated using the aluminum chloride (Sigma-Aldrich, MA, USA) colorimetric method [41,43]. A standard solution (1 mg aluminum chloride/mL) was prepared by dissolving HPLC grade quercetin dihydrate (HWI ANALYTIK GMBH, Rülzheim, Germany) in 80% ethanol (Sigma-Aldrich, MA, USA). The standard curve was generated [10]. The test solutions included a mixture of 20 µL of plant extracts with 60 µL of 95% ethanol, 4 µL of 10% aluminum chloride, 4 µL of 1 M potassium acetate, and 112 µL of distilled water (200 µL reaction mixture). These mixtures containing fresh and dry extracts were incubated at 25 °C for 30 min and measurements were taken at an absorbance of 415 nm. The samples were run in triplicate in a succession. The blank was prepared without the addition of aluminum chloride. The absorbance data was plotted against total flavonoid content using a standard curve generated by quercetin dehydrate (Sigma-Aldrich, Waltham, MA, USA).

***Antioxidant capacity measurement*.** Trolox equivalent antioxidant capacity (TEAC) assay of a sample was calculated based on the inhibition of radical cation absorption exerted by the standard TROLOX solution (6-hydroxy-2,5,7,8-tetramethy-chroman-2-carboxylic acid) (Sigma-Aldrich, MA, USA), a vitamin E analog [44]. A 7 mM ABTS solution [2,2′-azinobis(3-ethylbenzothiazo-line-6-sulfonic acid) diammonium salt] (Sigma-Aldrich, MA, USA) was mixed with 6.6 mg of potassium persulfate (Sigma-Aldrich, MA, USA) to make a final concentration of 2.45 mM. This ABTS radical solution was incubated in the dark at 25 °C for 16 h, and then diluted with ethanol to get an optical density (OD) A734 = 0.70 ± 0.02 with spectrophotometer in Multiskan™ GO Microplate using a polystyrene 96 well microplate (Costar #3628, Corning Inc., New York, NY, USA). Once the OD A734 = 0.70 ± 0.02 was achieved, 180 μL of ABTS was measured again. To the same measured ABTS, 20 μL of a *S. havanensis* extract was added. The measurement was conducted again after 6 min for the mixture. The measurements from all extracts were plotted against Trolox standards for percent inhibition at 6 min. Calculation of antioxidant capacity was expressed as percent inhibition according to the equation:% Inhibition = [(A_Control_ − A_Sample_)/A_Control_] × 100
where A_Control_ is the absorbance of the control reaction (containing all reagents except the test compound), A_Sample_ is the absorbance of the test compound, and % inhibition is the inhibition of ABTS absorbance by TROLOX.

### 2.4. Culture of HCT 116 Cell Line

The human colorectal carcinoma cell line HCT 116 was obtained from ATCC^®^ (Number: CCL-247™) and was cultured in the McCoy’s medium which had been supplemented with FBS (10%, *v*/*v*) and penicillin-streptomycin solution (1:1000) provided by ATCC^®^ (Number: 30-2300). The cells (5 × 103) were seeded, in triplicate, into the 96 well plates and incubated under 5% CO_2_ atmosphere at 37 °C for 24 h. Then the various dilutions of methanolic fresh and dry leaf extracts and control (without the extract), 1/2, 1/10, 1/100, and 1/1000), were added to the cells for incubation periods of 3, 24, 48, and 72 h at 37 °C respectively.

### 2.5. Cell Viability by MTT Colorimetric Method

Cell viability was assessed by using a 3-(4,5-dimethyl-thiazol-2-yl)-2,5-diphenyltetrazolium bromide (MTT) (Sigma-Aldrich, St. Louis, MO, USA) -based colorimetric assay. Cells in 96 well plates (5000 cells/well) were exposed to various dilutions of fresh and dry extracts (0 as control, 100%, 1/10, 1/100, and 1/1000), then incubated under 5% CO_2_ enriched atmosphere at 37 °C for each incubation period. The 30 μL MTT solution (5 mg/mL in phosphate-buffered saline) was added and further incubated for 4 h at 37 °C. After aspirating the supernatant from the wells, 200 μL dimethyl sulfoxide (DMSO) were added to dissolve formazan crystals. The absorbance of each well was observed at 595 nm using the Multiskan™ GO Spectrophotometer (Thermo Fisher Scientific, Waltham, MA, USA). Finally, these results were confirmed by the Trypan Blue dye exclusion method [45] using 1:1 cell/dye dilution, and results processed by the Countess^®^ II FL Automated Cell Counter (Life Technologies, Thermo Fisher Scientific, Waltham, MA, USA). The percentage of inhibition was calculated using the following formula:$$\frac{\%\;\mathrm{Inhibition}\;=100-\left(\;\mathrm{mean}\;\mathrm{absorbance}\;\mathrm{of}\;\mathrm{extract}\right)\times 100\;}{\mathrm{Mean}\;\mathrm{absorbance}\;\mathrm{of}\;\mathrm{positive}\;\mathrm{control}}$$
% Viability=100− % Inhibition

**Data collection and Statistical analysis:** All data are presented as means ± SE for at least three replications for each sample. Statistical analysis was based on two-way analysis of variance (ANOVA) and Tukey’s post hoc mean separation test with results at *p* ≤ 0.05 level was considered not statistically significant.

## 3. Results and Discussion

### 3.1. Micropropagation

Among the various BAP treatments, 10 µM BAP treatment with 10 day-long incubation produced the highest number of adventitious shoots using nodal explants (Figure 1A; Table 1) upon transfer to elongation medium for 21 days (Figure 1B,C). After incubating cultures for various time points in the shoot induction treatments, cultures were transferred to MS basal elongation medium for 21 days to allow adventitious buds to elongate. Control group exhibited bud break with possible elongation of two axillary buds. As there were no adventitious shoots produced in the control treatment, their number was kept zero across the board (Table 1). Elongated shoots rooted in MS basal medium and MS medium supplemented with IBA both, acclimatized and then finally transferred to pots containing potting soil and moved to the greenhouse (Figure 1D, E). Similar reports are available in Scutellaria species experimented for micropropagation [27,28,30]. A prolific shoot induction protocol was optimized for *Scutellaria ocmulgee* using transverse thin cell layer cultures (tTCL) of leaf and stem explants which was helpful in generating transgenic plants [33]. In this study, both cytokinins, BAP and TDZ, were effective in producing a high number of shoots.

### 3.2. Optimization of Agroinfection and Co-Cultivation

Transient GFP expression was observed at the cut ends and wounded sites of the nodal, internodal segments, and leaf segments. Transient GFP expression was observed in explants after 3 days of co-cultivation (Figure 1F–H). A total of 78 replicates for each type of explants were used for five bacterial density treatments. In this preliminary experiment, internode explants registered the highest infectivity on the basis of a number of explants expressing transient GFP expression (50%) followed by nodes (35.89%) and petioles (20.5%). It was interesting to note that OD_600_ = 1 of bacterial suspension was the most effective treatment for all three types of explants recording 70% of internode and nodal and 50% of petioles expressing GFP. Another treatment OD_600_ = 0.6 was the next most effective treatment as it resulted in 50, 40, and 20% infection and GFP expression in internode, node, and petiole explant, respectively. After co-cultivation, the explants were transferred to shoot induction medium containing 250 mg/L of antibiotics Carbenicillin and cefotaxime to eliminate *Agrobacterium*. Explants were screened for bacterial contamination and positive transient GFP expression daily. Thirty percent of the original explants retained transient GFP expression after 5 days. Leaf explants were difficult to maintain in the long run as they succumbed to *Agrobacterium* overgrowth. It is clear from the ultrastructure of the leaf that due to the extensive network of trichomes, it is difficult to eradicate bacteria. Transient expression was evident at the cut ends of the nodal and internodal explants, especially when callusing occurred. More studies are needed to regenerate transgenic plants.

**Total polyphenol content:** The gallic acid standard curve showed a linear correlation between absorbance and gallic acid equivalents with R^2^ = 0.9862, by the Folin–Ciocalteu reagent method. Fresh and dried leaf extracts of *S. havanensis* showed differences in the total polyphenol content (Figure 2A). In fresh extracts, TPP results ranged from 45.64 to 59.77 mg/g gallic acid equivalent (GAE). In dry leaf extracts, TPP contents were lower and ranged from 30.96 to 47.43 mg/g GAE.

**Flavonoid content:** Standard curve to determine total flavonoid content by using, as reference, quercetin, was linear (R^2^ = 0.9961). Total flavonoid content of *S. havanensis* ranged between 129.16 μg/mL to 155.37 μg/mL in fresh leaf extracts. In dried leaf extracts, flavonoid content ranged from 103.74 μg/mL to 135.77 μg/mL. The higher flavonoid content was observed in the fresh leaf extract (Figure 2B).

**Antioxidant capacity measurement:** Antioxidant capacity of fresh and dry leaf extracts of *S. havanensis* was measured at 1:10 dilution using the Trolox Equivalent Antioxidant Capacity (TEAC) assay. The standard curve showed Trolox percent inhibition of ABTS radicals generated by persulfate oxidation (R^2^ = 0.9986). A higher TEAC (Figure 2C) was obtained in fresh extract (33.68%) as compared to dry extract (15.48%). Figure 2 summarizes total polyphenol, flavonoid content, and antioxidant capacity of dry and fresh extracts of *S. havanensis*. In all cases, the concentration of phytochemicals is higher in the extracts derived from the fresh rather than the dry leaves.

Fresh and dry leaf extracts of the *S. havanensis* showed differences in the total polyphenol content. In fresh extracts, the TPP average result was 51.46 mg/g GAE and for dry extracts 41.23 mg/g GAE. However, the higher TPP content was recorded in the fresh extract (Figure 2A). A similar study on sixteen *Scutellaria* species exhibited higher polyphenol and flavonoid content in the fresh leaf extracts of 10 and 14 species respectively when compared with the extract obtained from dried leaves [10]. The difference in polyphenol and flavonoid content between dry and fresh extracts could be associated with the thermal stability of these compounds. An increase in polyphenol decomposition and volatilization has been observed in plant extracts subjected to a higher temperature [46]. Total flavonoid content of *S. havanensis* ranged between 129.162 μg/mL to 155.37 μg/mL in fresh leaf extracts. In dry leaf extracts, flavonoid content ranged from 103.74 μg/mL to 135.77 μg/mL, suggesting various *Scutellaria* species depend on the thermostability of flavonoid groups during leaf drying and extract preparation [47].

A higher antioxidant capacity on the basis of TEAC assay was observed in the fresh leaf extracts (Figure 2C). Previous research on *S. havanensis* confirmed that fresh herb extract shows a higher concentration of flavonoids than the dried herb, possibly due to higher temperature and thermal instability of the flavonoids [48]. Comparing the results of this study with those of for various *Scutellaria* species [10], we conclude, that fresh extracts of *S. havanensis* showed a higher TPP content than the dry extract of *S. albida*. Furthermore, the flavonoid content of both fresh and dry extracts of *S. havanensis* was higher than in other *Scutellaria* species, such as *S. albida*, *S. altissima*, *S. costaricana*, *S. drummondi*, *S. elliptica*, *S. incana*, *S. integrifolia*, *S. lateriflora*, *S. ovata*, *S. scandens* and *S. suffrutescens*. *S. havanensis* shows a higher flavonoid content of the fresh extract when compared to *Rosmarinus officinalis*, which is a common herb with well-known high antioxidant potential that has been used as a standard [49]. The inhibition percent for both fresh and dried extracts was higher in *S. havanensis* when compared with *S. drummondii*., *S. havanensis* extracts showed more inhibition than *S. lateriflora*, *S. albida*, *S. elliptica*, *S. incana*, *S. integrifolia,* and *S. ovata* [10].

**Fluorescent detection of glandular trichomes.** DPBA staining clearly differentiated non-glandular and glandular trichomes (Figure 1I,J). It is clear that non-glandular trichomes and stalks of glandular trichomes do not fluoresce when exposed to UV light. Secretory apical cells that contain secondary metabolites register a green, yellow, or orange image.

**Scanning electron microscopy.** The scanning electron micrographs reveal different types of trichomes on both adaxial and abaxial leaf surfaces (Figure 1K,L). The abaxial leaf surface has a higher density of hirsute capitate trichomes, though large peltate glandular trichomes are also visible (Figure 1K). The adaxial surface contains multicellular disc-like glandular peltate trichomes as well as multicellular capitate trichomes without glandular secretory vesicles (Figure 1L).

**HCT 116 cell line viability.** Cell viability was reduced when HCT 116 cells were incubated with *S. havanensis* extracts. Viability of HCT 116 cells was measured at different dilutions of *S. havanensis* dry and fresh extracts, during 3, 24, 48, and 72 h incubation periods and normalized with the viability of control cells. Significant difference analyzed by ANOVA (*p* < 0.05) in cell viability was obtained when using 1/100 and 1/1000 of dilutions of either dried or fresh extracts. Best results were obtained with 1/100 dilution (Figure 3). Percentage inhibition of *S. havanensis* extracts on the growth of the cell line HCT 116 was observed as measured by the MTT assay. Percentage inhibition for dry and fresh extracts was 90% and 87% respectively (Figure 3).

Dry and fresh extracts of *S. havanensis* decreased the viability of the colorectal cancer cell line HCT 116. A significant decrease in cell viability (ANOVA, *p* < 0.05) was observed in dry and fresh leaf extracts, as compared with control cells. Consistent results were obtained with plant extracts diluted to 1:100. These findings suggest that both fresh and dry extracts of *S. havanensis* may contain anticancer phytochemicals that inhibit cell growth. These results are consistent with previous research, in which flavonoids and phenolics have been reported as a free radical scavenger and as an inducer of apoptosis in leukemia, lung cancer, and colon adenocarcinoma cell lines [50]. Previous studies showed that quercetin has anticancer activities and was able to inhibit cancer cell growth of malignant mesothelioma cells in vitro [51]. Baicalein, which is another flavonoid present in *Scutellaria* species, at doses that are toxic to malignant cells [13] has shown no effect on the viability of normal human prostate epithelial cells [52]. Remarkably, wogonin treatment did not affect the viability of normal human prostate epithelial cells [53]. No significant difference in cell viability was observed when using either dry or fresh extracts. Both extracts showed percent inhibition values greater than 70% after 3 h. These results suggest that the potential anticancer substance is not affected by the slow drying procedure used in this work.

## 4. Conclusions

Predominant secondary metabolites from plants like polyphenols and flavonoids have been studied for their potential use as therapeutic compounds. They have shown considerable antioxidant activity; inhibition of cancer cell growth; induction of apoptosis, etc. Increasing populations, urbanization, and deforestation, mostly anthropogenic activities, are responsible for the endangered and extinct status of many valuable plants. In vitro multiplication can help assist germplasm conservation. This study optimized in vitro protocols for the efficient multiplication of plants that can be utilized for the isolation of therapeutic compounds or can be scaled up to produce clean biomass for the herbal supplement industry. Fresh and dried plant extracts both exhibited considerable antioxidant activity and when tested on cancer cell line HCT 116, dose-dependent inhibition was evident.

## Figures and Tables

**Figure 1 molecules-26-05813-f001:**
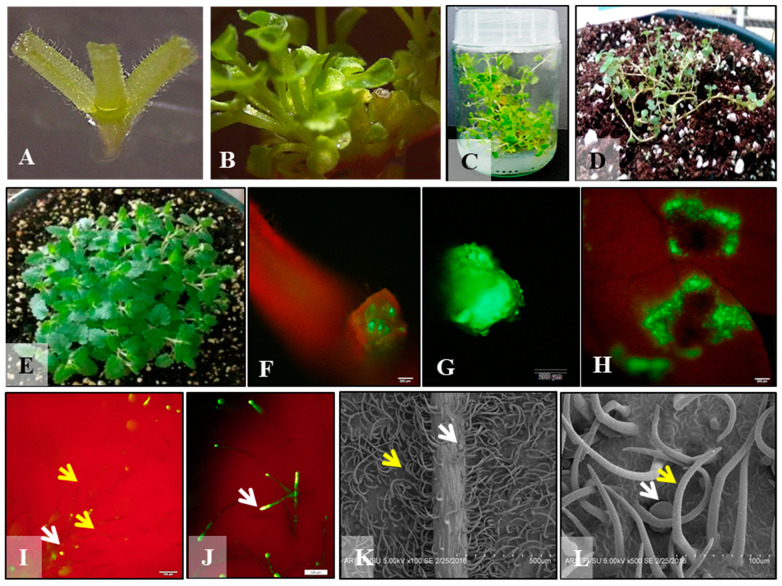
A nodal explant (**A**), A nodal explant in elongation medium with multiple shoots after 10 days incubation on MS medium + 10 µM BAP (**B**), Fully grown, elongated plantlets on MS basal medium (**C**), Ex vitro plantlet in a potting medium under intermittent mist in the greenhouse (**D**). Hardened plant in the greenhouse (**E**), GFP expression on the node, at the cut end of an internode, and wounded areas on *S. havanensis* leaf (**F**–**H**). Fluorescent staining differentiating non-glandular and glandular trichomes (**I**), Fluorescent detection of secondary metabolites in glandular trichomes (**J**), Scanning electron micrograph of abaxial leaf surface showing dense non-glandular trichomes (**K**). Glandular and non-glandular trichomes on the adaxial leaf surface. Yellow arrow: Non-glandular trichomes; White arrow: Glandular trichomes (**L**).

**Figure 2 molecules-26-05813-f002:**
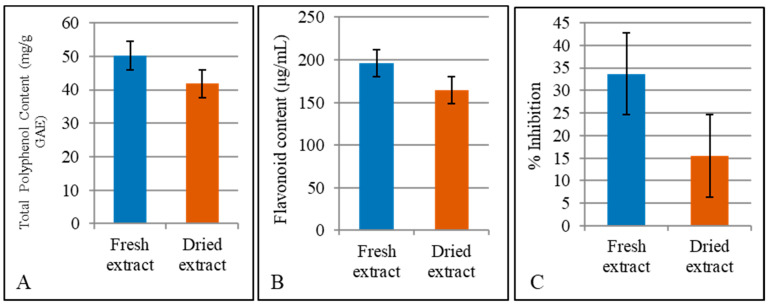
Bioactive compounds and antioxidant activity in the fresh and dry leaf extracts of *S. havanensis*. (**A**). Total Polyphenol Content of fresh and dried extracts expressed as mg/g of gallic acid equivalent (GAE mg/g) (**B**). Estimation of Total Flavonoid Content of expressed as µg/Ml (**C**). TROLOX Equivalent Antioxidant Capacity (TEAC) assay of dry and fresh extracts as percent inhibition.

**Figure 3 molecules-26-05813-f003:**
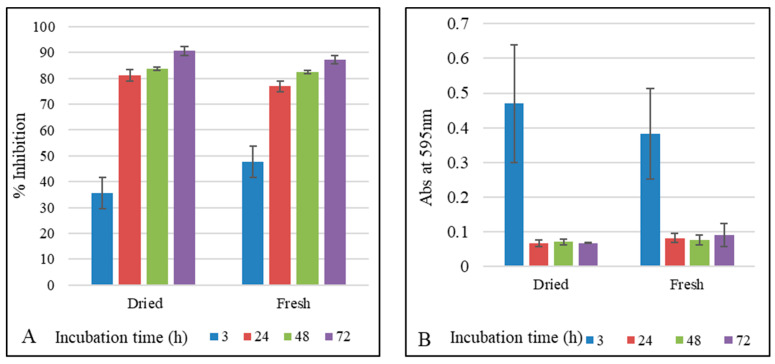
(**A**) Inhibition percentage of HCT 116 Cell line in the presence of 1:100 dilutions of dry and fresh extracts at 3, 24, 48, and 72 h incubation times. (**B**) Viability of HCT 116 cells using MTT assay in the presence of 1:100 dilution of dry and fresh leaf extracts after 3, 24, 48, and 72 h of incubation.

**Table 1 molecules-26-05813-t001:** Adventitious shoot bud induction in the nodal explants of *S. havanensis* in response to the various concentrations of cytokinin BAP.

Treatment	Number of Days in the Shoot Induction Medium and Number of Shoots				
	3	5	7	10	14
MS (Control)	Number of elongated adventitious shoots				
	0	0	0	0	0
MS + 0.05 µM BAP	2.67 ± 1.25 ^c^	5.33 ± 1.25 ^c^	2.67 ± 0.94 ^c^	2.67 ± 0.47 ^c^	2.67 ± 1.25 ^cd^
MS + 0.5 µM BAP	2.33 ± 0.94 ^d^	0.0	4.0 ± 2.83 ^c^	9.33 ± 4.78 ^b^	3.67 ± 1.89 ^c^
MS + 5.0 µM BAP	0.67 ± 0.94 ^d^	3.33 ± 2.87 ^c^	4.33 ± 0.47 ^c^	4.0 ± 2.94 ^c^	5.33 ± 1.25 ^c^
MS + 10.0 µM BAP	5.0 ± 0.82 ^c^	2.0 ± 0.82 ^d^	8.0 ± 4.55 ^bc^	14.67 ± 1.25 ^a^	* 1.0 ± 0.82 ^d^

* Hyperhydric cultures. Statistical analysis was carried out using analysis of variance (ANOVA) for the top three explants in each treatment. The means between 21 different treatments were analyzed using Tukey’s post hoc mean separation test and the treatments with the same letters are not significantly different (*p* < 0.05) level when compared at each induction period for different BAP concentrations. The control treatments contained no plant growth regulators.

## Data Availability

Data presented in the manuscript.

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
