# Peer review of "Micropropagation of Rare *Scutellaria havanensis* Jacq. and Preliminary Studies on Antioxidant Capacity and Anti-Cancer Potential"

_molecules, 2021, doi:10.3390/molecules26195813_

Round 1
Reviewer 1 Report
The article presents scientific interest with results consistent with the methodology. The manuscript needs minor revisions for publication.
The authors should make the use of genetic transformation more evident in relation to the rest of the work.
In the Introduction section, authors must justify the use of genetic transformation in the work context. How important is it to demonstrate GFP gene expression in relation to micropropagation and antioxidant and anticancer properties?
Likewise, this section in the results appears disconnected from the rest of the work. To the reader, it seems to be a combination of different works in this species, but that have no connection between them.
If this demonstration of genetic transformation is not useful or is not linked to in vitro culture, production of metabolites with biological properties, it should be removed from the manuscript.
The information " Prior to autoclaving the shoot induction medium, 30 g L-1 sucrose and 4 g L-1 gelzan (PhytoTechnology, USA) was added and pH was adjusted to 5.8-5.9" is repeated in the methodology.
Reviewer 2 Report
This manuscript deals with the micropropagation of Scutellaria havanensis and its evaluation as a source of antioxidants.
The manuscript could be interesting and seems to be reasonably accurate.
My main criticisms are the following:
1. The results of cell viability/inhibition assays should be reported for all extract concentrations, not just for the 1:100 dilution.
2. The results of biochemical assays are reported in a repetitive way in the text and section titles are incorrectly formatted, leading to some confusion.
3. The “antixidant capacity measurements” section cites Table 1, which however deals with shoot bud induction. No table is included on antioxidant capacity measurements.
In my opinion, the manuscript can be accepted after minor revision.
